

# Learning with semantic ambiguity for unbiased scene graph generation

Shanjin Zhong[1], Yang Cao[2], Qiaosen Chen[2] and Jie Gong[2]

[1] School of Artificial Intelligence, South China Normal University, Foshan, Guangdong, China
[2] School of Computer Science, South China Normal University, Guangzhou, Guangdong, China

## ABSTRACT

Scene graph generation (SGG) aims to identify and extract objects from images and elucidate their interrelations. This task faces two primary challenges. Firstly, the long-tail distribution of relation categories causes SGG models to favor high-frequency relations, such as "*on*" and "*in*". Secondly, some subject-object pairs may have multiple reasonable relations, which often possess a certain degree of semantic similarity. However, the use of one-hot ground-truth relation labels does not effectively represent the semantic similarities and distinctions among relations. In response to these challenges, we propose a model-agnostic method named Mixup and Balanced Relation Learning (MBRL). This method assigns soft labels to samples exhibiting semantic ambiguities and optimizes model training by adjusting the loss weights for fine-grained and low-frequency relation samples. Its model-agnostic design facilitates seamless integration with diverse SGG models, enhancing their performance across various relation categories. Our approach is evaluated on widely-used datasets, including Visual Genome and Generalized Question Answering, both with over 100,000 images, providing rich visual contexts for scene graph model evaluation. Experimental results show that our method outperforms state-of-the-art approaches on multiple scene graph generation tasks, demonstrating significant improvements in both relation prediction accuracy and the handling of imbalanced data distributions.

## INTRODUCTION

As computer vision technology progresses, people are no longer content with merely detecting and recognizing objects within images. Instead, there is a growing desire for a deeper level of understanding and reasoning about visual scenes. For example, when presented with an image, it is desirable not only to identify the objects present but also to generate textual descriptions based on the image content (image captioning) (*Yang et al., 2019*; *Gu et al., 2019*) and to find similar images (image retrieval) (*Johnson et al., 2015*; *Wang et al., 2020*; *Wei et al., 2022*). Additionally, machines may be expected to explain what actions are being performed in the image, such as what a little girl is doing (Visual Question Answering) (*Antol et al., 2015*; *Teney, Liu & van Den Hengel, 2017*; *Xiao et al., 2022*; *Li et al., 2022b*). Achieving these tasks requires a more advanced level of understanding and reasoning in image processing. Scene graphs are precisely such powerful tools for scene understanding. A scene graph provides a structured

Corresponding author
Yang Cao, yangcao@scnu.edu.cn

representation of an image by identifying objects (*e.g.*, "man", "bike") as nodes and their relations (*e.g.*, "riding") as edges. At present, research related to scene graph generation (SGG) (*Johnson et al., 2015*) is increasing rapidly. The SGG task can be divided into two subtasks: (1) Object detection and classification: Identifying objects in the image and assigning them to the correct categories; (2) relation prediction: Predicting the relations between pairs of detected objects.

However, current SGG methods face two main challenges: long-tail distribution (*Reed, 2001*) and semantic ambiguity (*Yang et al., 2021*).

Long-tail distribution signifies that a small number of relations account for the majority of samples, whereas a vast array of relations constitute only a minor portion of the dataset. As shown in Fig. 1A, relations such as "on" and "in" appear tens of thousands of times in Visual Genome (*Krishna et al., 2017*), whereas others like "*laying on*" and "*growing on*" appear merely a few hundred times. As a result, model predictions often favor high-frequency relations, many of which are trivial and offer limited informational value (*e.g.*, "on", "in").

Semantic ambiguity signifies that many samples can be described as either general relation category (*e.g.*, "on") or an informative one (*e.g.*, "*walking on*"). Although these relations are semantically close, their specific meanings vary. As illustrated in Fig. 1B, the relation between "*dog*" and "*sidewalk*" can be described by "*on*" as well as "*walking on*". Both relations involve one object being above another, hence they are semantically similar. However, "*walking on*" implies an act of movement, whereas "*on*" merely denotes a position in relation to something else without suggesting any movement. Therefore, accurately identifying and distinguishing these subtle semantic differences is crucial for generating accurate scene graphs.

To address the aforementioned challenges, existing unbiased SGG strategies can be broadly categorized into four main methods: (1) Re-sampling (*Dong et al., 2022*; *Li et al., 2021*): This method involves sampling additional training samples from low-frequency relations to balance the data distribution. (2) Re-weighting (*Yu et al., 2020*; *Yan et al., 2020*): This method focuses on enhancing the impact of low-frequency relation training samples in the loss calculation through various weighting strategies. (3) Biased-model-based (*Tang et al., 2020*; *Chiou et al., 2021*): This method aims to distinguish unbiased predictions within models that have been trained on biased data. (4) Data transfer (*Zhang et al., 2022*; *Li et al., 2022a*): This method involves transferring high-frequency relations to low-frequency relations and reassigning fine-grained labels to mitigate the unbalanced distribution of relations. Although these strategies address the challenges of imbalanced relation distribution and semantic ambiguity to some extent, they inadvertently diminish accuracy in recognizing high-frequency relations. As a result, this significantly undermines the overall performance of the model. The primary cause of this phenomenon is that these strategies treat relation classification as a single-label task, utilizing one-hot vectors that inadequately capture the semantic similarities and differences among relations. This representation inadequately captures the semantic similarities and differences, limiting the SGG model's learning and reasoning capabilities in complex scenes.

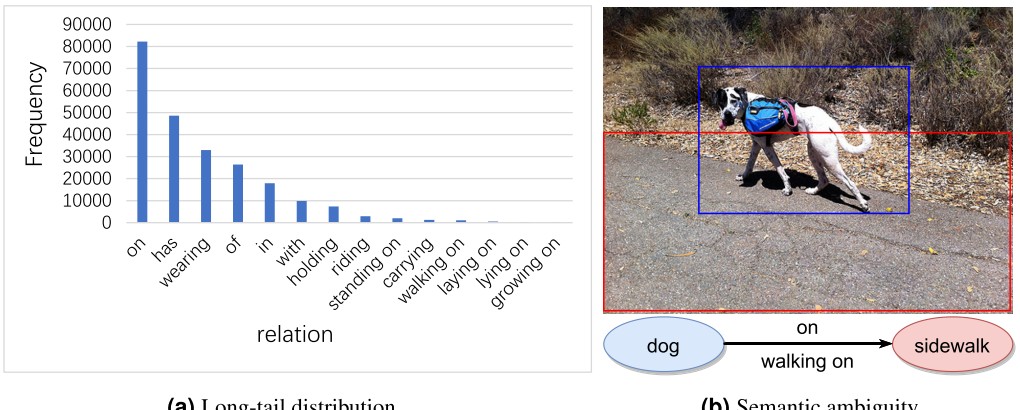

(a) Long-tail distribution      (b) Semantic ambiguity

**Figure 1** **Examples of long-tail distribution and semantic ambiguity in visual genome dataset.** Image credit: the Visual Genome dataset archive at https://homes.cs.washington.edu/~ranjay/visualgenome/.

To address the previously discussed challenges, we propose a novel framework in this article, termed Mixup and Balanced Relation Learning (MBRL), which can be seamlessly integrated into existing SGG models. This framework comprises two components: (1) Mixup relation learning (MRL) generates an enhanced dataset by merging semantically similar relations found in each subject-object pair into soft labels, thereby guiding the training process of the model. Unlike one-hot target labels, these soft labels provide a probabilistic distribution across potential relations. They reflect the degree of similarity and difference among the relations, allowing the model to more accurately address semantic ambiguities within the samples. (2) Balanced relation learning (BRL) discerns fine-grained relation samples using soft label scores and adjusts their weights accordingly. Simultaneously, BRL also adjusts the weights for those low-frequency relation samples that do not receive soft labels. Consequently, BRL not only improves the SGG model's capacity to discern fine-grained relations but also amplifies its focus on low-frequency relations, which are easily neglected. Through these strategies, MBRL reduces the impact of prediction errors and improves the SGG model's overall performance.

We evaluate our method using widely-used datasets: the Visual Genome dataset and the Generalized Question Answering dataset (*Hudson & Manning, 2019*). Given that MBRL is a model-agnostic debiasing strategy, it seamlessly integrates with various SGG models, thereby enhancing their performance. Extensive ablations and results on multiple SGG tasks and backbones have shown the effectiveness and generalization ability of MBRL.

In summary, our contributions are as follows:

(1) We introduce a novel model-agnostic method called MBRL, designed to assign soft labels to samples exhibiting semantic ambiguities, thereby enriching the dataset. Concurrently, MBRL enhances the efficacy of model training through adjusting the loss weights for both fine-grained and low-frequency relation samples.

(2) We conducted evaluations of our method using the Visual Genome and the Generalized Question Answering datasets, which significantly enhanced the performance

of benchmark models. The results demonstrate that MBRL can enable these models to achieve a satisfactory trade-off in performance between different relations.

# RELATED WORKS

## Scene graph generation

SGG is dedicated to transforming visual images into semantic graph structures, thereby playing a critical role in merging vision and language. Early methods such as VTransE (*Zhang et al., 2017*) focused on identifying objects and relations using separate networks, overlooking the wealth of contextual information. Subsequently, iterative message passing (IMP) (*Xu et al., 2017*) introduced an iterative message-passing mechanism to refine object and relation features, highlighting the substantial role contextual information plays in enhancing relation prediction accuracy. Motifs (*Zellers et al., 2018*) emphasizes the critical importance of contextual interplay among objects, utilizing BiLSTM to disseminate contextual data effectively. Similarly, Transformer (*Tang et al., 2020*) captures rich contextual representations of objects by encoding features through self-attention layers. To address the challenges posed by noisy information during message passing, VCTree (*Tang et al., 2019*) proposes a tree-structured method to efficiently leverage global contexts among objects. Additionally, KERN (*Chen et al., 2019*) attempts to incorporate prior knowledge into SGG models to improve the precision of relation predictions. Nonetheless, these methods overlook the long-tail distribution in data, resulting in a propensity for predictions to favor high-frequency relations. Such relations tend to be less informative, thereby constraining the utility of these models for downstream tasks.

## Unbiased scene graph generation

Unbiased scene graph generation methods aim to rectify the prediction biases stemming from the long-tail distribution of data, with a particular focus on enhancing the model's performance across various relations. They can be broadly classified into four categories: re-sampling (*Dong et al., 2022*; *Li et al., 2021*), re-weighting (*Yu et al., 2020*; *Yan et al., 2020*), biased-model-based (*Tang et al., 2020*; *Chiou et al., 2021*), and data transfer (*Zhang et al., 2022*; *Li et al., 2022a*). Stacked hybrid-attention and group collaborative learning (SHA+GCL) (*Dong et al., 2022*) employs a median re-sampling strategy, adjusting the sample rates to balance the training sets according to the median relation count within each classification space. Bipartite graph neural network (BGNN) (*Li et al., 2021*) utilizes a bi-level re-sampling method to achieve a balance in data distribution during the training phase. CogTree (*Yu et al., 2020*) leverages semantic relations across different categories to devise a loss function that rebalances the weights. Predicate-correlation perception learning (PGPL) (*Yan et al., 2020*) dynamically identifies appropriate loss weights by recognizing and leveraging relation category correlations. TDE (*Tang et al., 2020*) calculates the difference between the original and counterfactual scenes to remove context bias, ensuring unbiased scene graph generation. Dynamic label frequency estimation (DLFE) (*Chiou et al., 2021*) dynamically estimates label frequencies by maintaining a moving average of biased probabilities, allowing the model to recover unbiased probabilities.

Although these methods alleviate bias and improve low-frequency relations performance, they often compromise high-frequency relations performance and neglect the semantic ambiguity inherent in visual relations. Recent works (*Zhang et al., 2022*; *Li et al., 2022a*) argue that semantic ambiguity could be alleviated if there is a reasonable and sound dataset. IETrans (*Zhang et al., 2022*) introduces an Internal and External Data Transfer method to achieve the transfer of high-frequency to low-frequency relations and the relabeling of relations for unannotated samples. NoIsy label correction (NICE) (*Li et al., 2022a*) redefines SGG as a noisy label learning issue, presenting a strategy for noisy labels correction aimed at bias mitigation. It effectively cleanses noisy dataset annotations to equalize the data distribution.

These methods treat relation classification as a single-label problem and use one-hot target labels to train the relation classifier in SGG models. In one-hot target labels, each relation is represented as a binary vector where only one relation is set to 1 (indicating the target relation), and all other relations are set to 0. This method is highly effective for clear and mutually exclusive classification tasks. However, it fails to capture the nuances in scenes with semantic ambiguities, where relations are not mutually exclusive. In contrast, soft labels assign a probability to each relation, indicating the likelihood that the sample belongs to each relation and revealing the subtle differences between them. Our proposed method improves upon this by generating a training label distribution that considers semantic similarities and differences between relations. This method achieves balanced performance across both high-frequency and low-frequency relations in the model.

## Label smoothing and label confusion

Label smoothing (*Szegedy et al., 2016*) is a regularization technique designed to prevent overly confident predictions on training examples. It achieves this by mixing one-hot label vectors with a uniform noise distribution. However, this method of generating soft labels, primarily by introducing noise, fails to capture the semantic ambiguity within samples. Label confusion learning (*Guo et al., 2021*) was proposed for text classification tasks, introducing a label confusion model that calculates the similarity between instances and labels during training. This model generates a probability distribution, superseding the traditional one-hot label vectors. In addition, label semantic knowledge distillation (LS-KD) (*Li et al., 2023*) dynamically generates soft labels for each subject-object pair by merging the model's relation label prediction distribution with the original one-hot labels. However, the prevailing long-tail distribution skews the model's predictions towards more frequent relations, making it challenging to generate soft labels that accurately reflect the differences between relations. In contrast to these methods, we measure the similarity and differences between relations by calculating the amount of information for each relation. This method ensures that the generated soft labels more accurately reflect the similarities and differences between relations, leading to improved model performance and better handling of low-frequency relations.

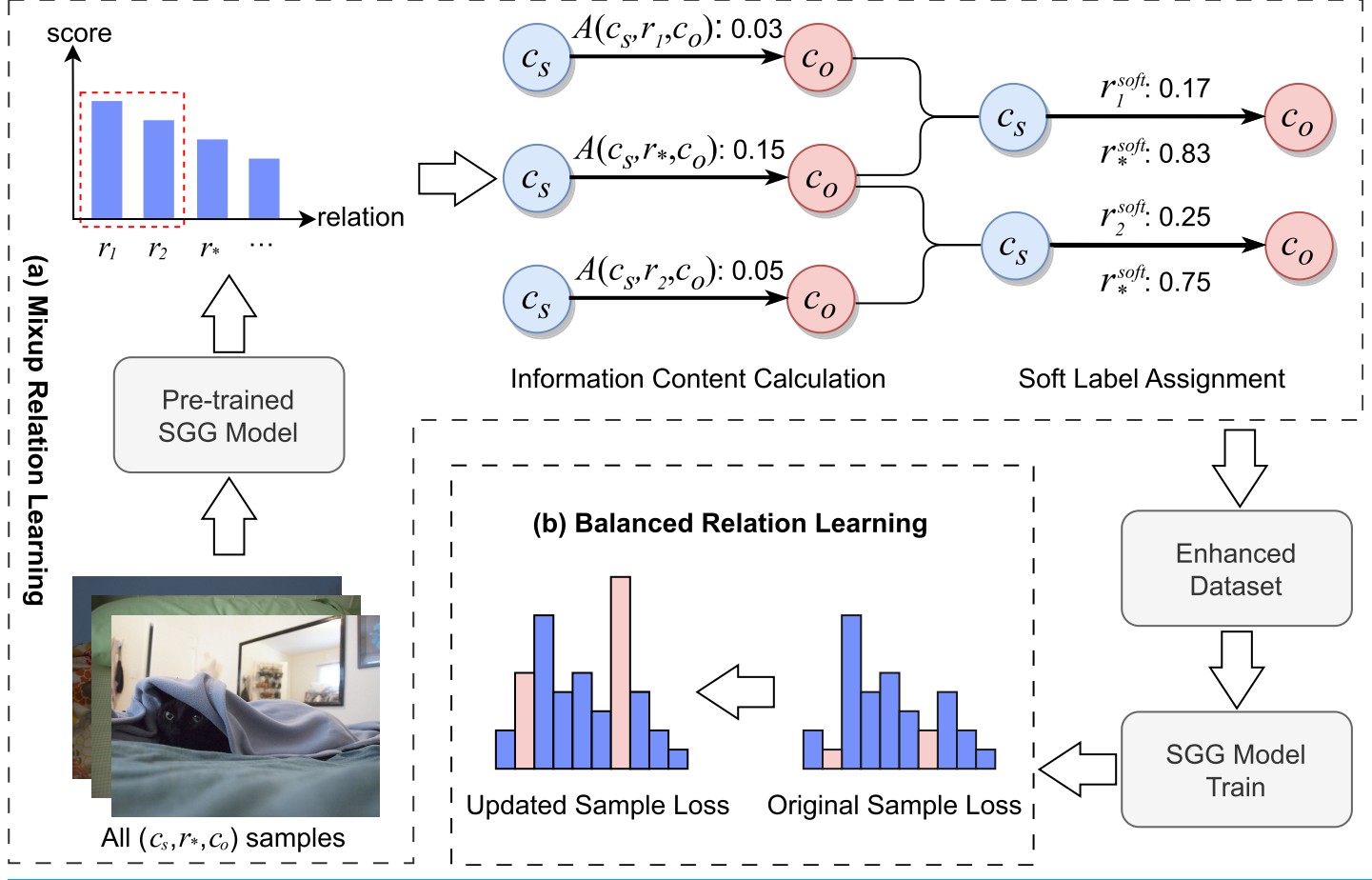

**Figure 2** **The pipeline of MBRL.** (A) MRL: for each relation triplet $(c_s, r_*, c_o)$, the MRL module identifies triplets with semantic similarities and assigns soft labels to them. (B) BRL: for all samples, the BRL module identifies fine-grained and low-frequency relation triplets, adjusting their loss weights accordingly. Image credit: the Visual Genome dataset archive at https://homes.cs.washington.edu/~ranjay/visualgenome/.

## METHOD

This section offers a detailed outline of our method. In standard SGG pipelines, objects are first detected, followed by the prediction of relations between them. Our proposed MBRL framework is specifically designed for the relation prediction stage.

Figure 2 illustrates the overall process of the MBRL framework. Initially, training samples are input into a pre-trained SGG model to obtain the relational probability distribution for each sample. Subsequently, for each category of relational triplets, the MRL module aggregates the relational probability distributions of corresponding samples and discerns relations that are semantically close to the ground-truth label. It then allocates soft labels to samples with the same subject-object pairs that exhibit relations semantically close to the ground-truth label. In this way, MRL generates an enhanced training dataset. Finally, the BRL module identifies fine-grained relation samples through soft label scores and modifies their loss weights during the training of the SGG model. It also adjusts the loss weights of low-frequency relation samples that have not been assigned soft labels.

## Problem definition

The task of SGG is to construct a scene graph $G$ for a given image $I \in \mathbb{R}^{H \times W \times 3}$. This graph $G$ comprises a set of objects $O = \{(b_i, c_i)\}_{i=1}^{N_o}$ and a set of relation triplets $E = \{(s_i, r_i, o_i)\}_{i=1}^{N_e}$, collectively denoted as $G = (O, E)$. Each object in $O$, represented by $(b_i, c_i)$, includes an object bounding box $b_i \in \mathbb{R}^4$ and an object category $c_i$, which is part of the pre-defined object category set $C$. Furthermore, each relational triplet $(s_i, r_i, o_i)$ is composed of a subject $s_i \in O$, an object $o_i \in O$, and a relation $r_i$ between them, where $r_i$ is a member of the predefined set of relation categories $R$.

## Mixup relation learning

To tackle semantic ambiguity, the Mixup relation learning (MRL) module enriches the dataset by allocating soft labels to samples of relation triplets that exhibit semantic ambiguities. These soft-labeled samples are subsequently employed in the training of SGG models.

Following *Zhang et al. (2022)*, we first identify confusion pairs as semantically similar relation pairs, since informative relation categories are easily confused with general ones. Specifically, for each relation triplet category $(c_s, r_*, c_o)$, we use a pre-trained baseline model to predict relation labels of all samples belonging to $(c_s, r_*, c_o)$ in the training set, and average their score vectors. Subsequently, relations with a predicted score higher than that of the ground-truth relation are regarded as semantically similar to the ground-truth relation $r_*$. This is formalized as $R_{sim} = \{r_i | p_{r_i} > p_{r_*}\}$, where $p_{r_i}$ is the predicted score for the $i$-th relation and $p_{r_*}$ denotes the predicted score for the ground truth relation $r_*$. Based on this, we collect all samples in the training set satisfying Eq. (1):

$$T_{sim} = \{(s_j, r_j, o_j) \mid (c_{s_j} = c_s) \wedge (r_j \in R_{sim}) \wedge (c_{o_j} = c_o)\} \tag{1}$$

where $\wedge$ denotes the logical conjunction operator. We quantify the information contained in $r_*$ and $r_j$ within the subject-object pair. Soft labels are then assigned to all samples in $T_{sim}$ based on the proportion of information content between $r_j$ and $r_*$, replacing the original one-hot labels $r_j$.

To achieve this, we use an attraction factor (*Zhang et al., 2022*) to calculate the amount of information contained in the relation within each relational triplet, as defined in Eq. (2):

$$A(c_s, r_*, c_o) = \frac{N(c_s, r_*, c_o)}{\sum_{c_i, c_j \in C} I(c_i, r_*, c_j) \cdot N(c_i, r_*, c_j)} \tag{2}$$

where $N(c_s, r_*, c_o)$ denotes the number of samples of the relation triplet $(c_s, r_*, c_o)$ within the training set, and $I(c_i, r_*, c_j)$ indicates whether the triplet category $(c_i, r_*, c_j)$ exists in the training set. $I(c_i, r_*, c_j)$ returns 1 if the relation triplet $(c_i, r_*, c_j)$ is present in the training set, and 0 otherwise. A higher $A(c_s, r_*, c_o)$ indicates that the relation triplet is relatively more unique or carries more information within the entire dataset. This is because it represents a larger proportion among all triplets with relation $r_*$. Based on this, we assign the relation $r_*$ from $(c_s, r_*, c_o)$ to each relation triplet in $T_{sim}$. Specifically, for each relation triplet $(s_j, r_j, o_j)$ in $T_{sim}$, we compute its semantic similarity to the target relation $r_*$ and

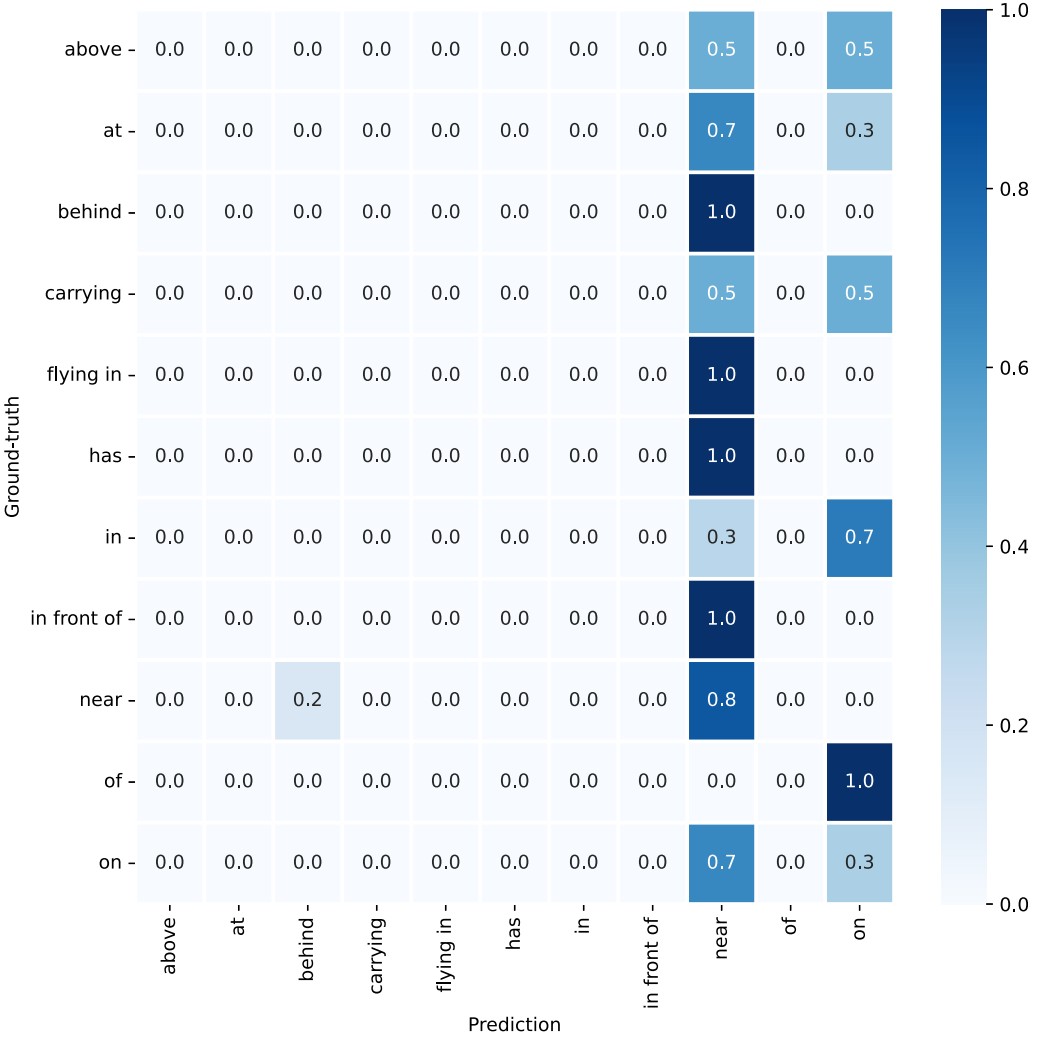

**Figure 3 Confusion matrix for the motifs model in the VG training set, featuring "*plane*" as both the subject and the object.**

generate the corresponding soft labels $r_j^{soft}$ and $r_*^{soft}$ by normalization. These two soft labels represent the similarity between $r_j$ and $r_*$, as defined in Eqs. (3) and (4):

$$r_j^{soft} = \frac{A(c_s, r_j, c_o)}{A(c_s, r_j, c_o) + A(c_s, r_*, c_o)} \tag{3}$$

$$r_*^{soft} = \frac{A(c_s, r_*, c_o)}{A(c_s, r_j, c_o) + A(c_s, r_*, c_o)} \tag{4}$$

The denominator represents the total amount of information contained in the two relations, $r_*$ and $r_j$, within the same subject-object context. The resulting quotient produces a score that falls within the range of 0 to 1, reflecting their semantic similarity and differences. Higher scores indicate greater similarity, while lower scores indicate significant differences. Next, soft labels $r_*^{soft}$ and $r_j^{soft}$ are assigned to all samples in $T_{sim}$. However, not all samples receive soft labels. As the confusion matrix in Fig. 3 shows, the relation "*flying*

*in*" is not incorrectly assigned to other categories. This indicates that "*flying in*" is distinctive enough to be clearly identifiable, thus making soft labeling unnecessary and potentially misleading for such unique cases.

## Balanced relation learning

In this module, our objective is to address the challenges presented by the long-tail distribution by modifying the loss weights for each fine-grained and low-frequency relation sample. Fine-grained relations usually offer more specific and detailed information than coarse-grained relations, thus possessing greater informational value in numerous contexts. To effectively differentiate between these two types of relations and utilize this distinction to improve model performance, we set a threshold $\theta$. Soft label scores that exceed $\theta$ are considered fine-grained relations. Upon classifying a relation as fine-grained, we adjust its loss weight by applying the loss balancing hyperparameter $\alpha$, ensuring that these relations receive appropriate attention and emphasis during the model training process. During training, we adjust the cross-entropy loss to accommodate soft label training, as defined in Eq. (5):

$$L_{soft} = -\sum_{i=1}^{N} w_i r_i^{soft} \log(p_i) \tag{5}$$

where $N$ denotes the total number of relation categories, $r_i^{soft}$ represents the score of the $i$-th relation category in the soft label, $p_i$ indicates the prediction probability of the $i$-th relation category, and $w_i$ is the weight assigned to each relation label. The weight $w_i$ as defined in Eq. (6):

$$w_i = \begin{cases} \alpha, & \text{if } r_i^{soft} \geq \theta \\ 1, & \text{otherwise} \end{cases}. \tag{6}$$

In model training, low-frequency relations that appear in only a small number of samples are often neglected, which can result in these relations receiving less emphasis during the learning process. Nevertheless, these low-frequency relations may carry unique and valuable information that contributes to the model's overall performance. In the MRL module, not all low-frequency relation samples are assigned soft labels. To ensure that all low-frequency relation samples are given adequate consideration during training, we apply the loss balancing hyperparameter $\alpha$ to adjust the loss weights for these single-label samples, as defined in Eq. (7):

$$L_{single} = -\alpha \sum_{i=1}^{N} \log(p_i) r_i \tag{7}$$

where $r$ adopts a one-hot representation, meaning $\sum_{i=1}^{N} r_i = 1$ and $r_i = 1$ for the correct relation category, which denotes the ground-truth relation.

In order to handle both soft-labeled and single-labeled samples effectively, we compute the total loss by combining the individual losses for each type of sample. The final total loss function as seen in Eq. (8):

$$L_{total} = \frac{-\sum_{m_{soft}=1}^{M_{soft}} \sum_{i=1}^{N} w_i r_i^{soft} \log(p_i^{(m_{soft})}) - \sum_{m_{single}=1}^{M_{single}} \left( \alpha \sum_{i=1}^{N} \log(p_i^{(m_{single})}) r_i^{(m_{single})} \right)}{M_{soft} + M_{single}}. \quad (8)$$

Here, $M_{soft}$ and $M_{single}$ represent the total number of soft-labeled and single-labeled samples, respectively.

# EXPERIMENTS

In this section, we describe the experimental framework, including datasets, tasks, evaluation metrics, and implementation details. The effectiveness and generalization ability of the proposed method are then demonstrated through comparisons with various baseline models across different SGG datasets. We follow with ablation studies to evaluate the impact of each component and discuss the choice of hyperparameters. Finally, visualizations illustrate the method's ability to enhance the model's accuracy.

## Experimental settings

### Visual Genome dataset

Experiments were conducted on the Visual Genome (VG) dataset, comprising 108k images, 75k objects, and 37k relations. Following previous work (*Li et al., 2021*; *Yu et al., 2020*; *Xu et al., 2017*; *Zellers et al., 2018*; *Tang et al., 2020*), the widely-used VG150 split (*Xu et al., 2017*) was selected, encompassing the most frequent 50 relation categories and 150 object categories. Additionally, based on *Li et al. (2021)*, relations were classified into three categories according to the number of samples in the training set: head (greater than 10k), body (0.5k to 10k), and tail (less than 0.5k). The VG150 dataset's allocation was 70% for training, 30% for testing, with 5k training images reserved for validation.

### Generalized Question Answering dataset

Another dataset utilized in our experiments is the Generalized Question Answering (GQA) dataset, designed for vision-language tasks and featuring over 3.8 million relation annotations across 1,704 object categories and 311 relation categories. We conducted experiments on the GQA200 split (*Dong et al., 2022*), which consists of the Top-200 object categories and Top-100 relation categories. Similarly to VG150, the GQA200 dataset's allocation was 70% for training, 30% for testing, with 5k training images reserved for validation.

### Tasks

Following previous work (*Xu et al., 2017*; *Zellers et al., 2018*; *Tang et al., 2020*), we evaluate our method on three conventional tasks: (1) Predicate classification (PredCls) predicts the relations between objects given their labels and bounding boxes. (2) Scene graph classification (SGCls) predicts object categories and the relations between them, given bounding boxes. (3) Scene graph detection (SGDet) predicts object categories and the relations between them, starting with detecting object bounding boxes in images. In our experiments, the MRL module utilizes a pre-trained SGG model from the PredCls task to generate an enhanced dataset. The SGG model is then trained on this enhanced dataset for

each of the three tasks (PredCls, SGCls, SGDet) separately, with the BRL module adjusting loss weights during training. This approach ensures that the improvements in relation prediction are carried over to all tasks.

### Metrics

Following previous works (*Li et al., 2021*; *Zhang et al., 2022*; *Li et al., 2022a*), we use Recall@K (R@K), mean Recall@K (mR@K), and a composite metric called mean as our evaluation metrics. R@K calculates the percentage of top-K confidently predicted relation triplets that match the ground-truth. The formula is defined as:

$$R@K = \frac{|G \cap X_K|}{|G|} \tag{9}$$

where $G$ represents the set of ground-truth triplets, and $X_K$ represents the top-K predicted triplets. This metric measures the percentage of ground-truth relations that are successfully retrieved in the top $K$ predictions. In contrast, mR@K calculates R@K for each individual relation category and subsequently computes the average R@K across all relation categories. The formula is defined as:

$$mR@K = \frac{1}{|R'|} \sum_{r \in R'} \frac{|G(r) \cap X_K(r)|}{|G(r)|} \tag{10}$$

where $R'$ is the subset of relation categories present in the ground truth triplets, $G(r)$ and $X_K(r)$ are the ground truth and predicted triplets for relation $r$, respectively. This metric ensures that rare relations are not overshadowed by common ones. However, optimizing based solely on mR@K may cause the model to overemphasize low-frequency relations while neglecting more prevalent relations. Though theoretically promoting a balanced performance distribution, this method may not accurately evaluate the model's ability to identify more common and essential real-world relation categories. Therefore, we adopt the mean metric, which averages the R@K and mR@K scores, to provide a more balanced evaluation of performance.

### Implementation details

Following previous work (*Dong et al., 2022*; *Li et al., 2021*; *Zhang et al., 2022*; *Tang et al., 2020*), we adopted a pre-trained Faster R-CNN with ResNeXt-101-FPN provided by *Tang et al. (2020)* as the object detector, which was trained on the VG dataset. For MBRL, parameters $\theta$ and $\alpha$ were empirically set to 0.95 and 5, respectively, after exhaustive experimentation demonstrated these values consistently yielded optimal performance outcomes. Table 1 shows the specific parameter settings. Other training settings follow (*Zhang et al., 2022*). All experiments are conducted on an A5000 GPU.

## Compared methods

To prove its performance, we compare it with state-of-the-art methods. These include classic feature- and relation-based models like motifs (*Zellers et al., 2018*) and VTransE (*Zhang et al., 2017*), more structurally complex approaches like Transformer (*Tang et al., 2020*) and VCTree (*Tang et al., 2019*), and knowledge-augmented models such as KERN

**Table 1  Experimental settings for object detectors and SGG models.**

| Model | Dataset | Batch size | Learning rate | Optimizer | Momentum | Additional parameters |
|---|---|---|---|---|---|---|
| Faster R-CNN with ResNeXt-101-FPN | GQA | 8 | $8 \times 10^{-3}$ | SGD | 0.9 | |
| Faster R-CNN with VGG16 | VG | 8 | $8 \times 10^{-3}$ | SGD | 0.9 | |
| Motifs, VCTree | VG, GQA | 12 | 0.12 | SGD | 0.9 | Faster R-CNN with ResNeXt-101-FPN |
| Motifs, VCTree | VG | 12 | 0.012 | SGD | 0.9 | Faster R-CNN with VGG16 |
| Transformer | VG, GQA | 16 | 0.008 | SGD | 0.9 | Faster R-CNN with ResNeXt-101-FPN |
| Transformer | VG | 16 | 0.008 | SGD | 0.9 | Faster R-CNN with VGG16 |

**Table 2  Performance (%) comparison of different methods on the VG150 dataset.** Bold entries indicate the best results.

| Model | PredCls | | | SGCls | | | SGdet | | |
|---|---|---|---|---|---|---|---|---|---|
| | R@50/100 | mR@50/100 | Mean | R@50/100 | mR@50/100 | Mean | R@50/100 | mR@50/100 | Mean |
| BGNN | 59.2/61.3 | 30.4/32.9 | 46.0 | 37.4/38.5 | 14.3/16.5 | 26.7 | 31.0/35.8 | 10.7/12.6 | 22.5 |
| PCPL | 50.8/52.6 | 35.2/37.8 | 44.1 | 27.6/28.4 | 18.6/19.6 | 23.6 | 14.6/18.6 | 9.5/11.7 | 13.6 |
| VTransE | 65.7/67.6 | 14.7/15.8 | 41.0 | 38.6/39.4 | 8.2/8.7 | 23.7 | 29.7/34.3 | 5.0/6.1 | 18.8 |
| KERN | 65.8/67.6 | 17.7/19.2 | 42.6 | 36.7/37.4 | 9.4/10.0 | 23.4 | 27.1/29.8 | 6.4/7.3 | 17.7 |
| SHA+GCL | 35.1/37.2 | 41.6/44.1 | 39.5 | 22.8/23.9 | 23.0/24.3 | 23.5 | 14.9/18.2 | 17.9/20.9 | 18.0 |
| Motifs | 64.9/66.9 | 15.0/16.4 | 40.8 | 38.0/38.9 | 8.7/9.3 | 23.7 | 31.0/35.1 | 6.7/7.7 | 20.1 |
| +GCL | 42.7/44.4 | 36.1/38.2 | 40.4 | 26.1/27.1 | 20.8/21.8 | 24.0 | 18.4/22.0 | 16.8/19.3 | 19.1 |
| +CogTree | 35.6/36.8 | 26.4/29.0 | 32.0 | 21.6/22.2 | 14.9/16.1 | 18.7 | 20.0/22.1 | 10.4/11.8 | 16.1 |
| +IETrans | 53.0/55.0 | 30.3/33.9 | 43.1 | 32.9/33.8 | 16.5/18.1 | 25.3 | 25.4/29.3 | 11.5/14.0 | 20.1 |
| +NICE | 55.1/57.2 | 29.9/32.3 | 43.6 | 33.1/34.0 | 16.6/17.9 | 25.4 | 27.8/31.8 | 12.2/14.4 | 21.6 |
| +DLFE | 52.5/54.2 | 26.9/28.8 | 40.6 | 32.3/33.1 | 15.2/15.9 | 24.1 | 25.4/29.4 | 11.7/13.8 | 20.1 |
| +TDE | 46.2/51.4 | 25.5/29.1 | 38.1 | 27.7/29.9 | 13.1/14.9 | 21.4 | 16.9/20.3 | 8.2/9.8 | 13.8 |
| **+MBRL** | 56.4/58.3 | 33.7/37.2 | **46.4** | 33.6/34.4 | **19.7/21.4** | **27.3** | 27.2/31.5 | 13.3/16.1 | **22.0** |

(*Chen et al., 2019*). Additionally, we evaluate against recent unbiased SGG methods, including SHA+GCL (*Dong et al., 2022*), BGNN (*Li et al., 2021*), and PCPL (*Yan et al., 2020*), which aim to address data bias challenges and improve generalization. Given the model-agnostic nature of our framework, we further compare it with other model-agnostic methods like group collaborative learning (GCL) (*Dong et al., 2022*), total direct effect (TDE) (*Tang et al., 2020*), DLFE (*Tang et al., 2020*), CogTree (*Yu et al., 2020*), IETrans (*Zhang et al., 2022*), and NICE (*Li et al., 2022a*), to illustrate its seamless integration capability and performance improvements.

## Comparison with state-of-the-art methods
### VG150

Table 2 shows the comparison results of motifs combined with our MBRL. From the results, the enhancements in mR@K and mean metrics demonstrate that our method improves the model's capacity to identify a broader range of relations. While MBRL shows a reduction in the R@100 metric (from 66.9 to 58.3 for PredCls), the decrease can be

**Table 3 Performance (%) of our method applied to three different baseline models with various object detector backbones for the PredCls task on the VG150 dataset.** Bold entries indicate the best results.

| Backbone | SGG model | PredCls | | | | | |
| --- | --- | --- | --- | --- | --- | --- | --- |
| | | R@50/100 | mR@50/100 | Mean | Head mR@100 | Body mR@100 | Tail mR@100 |
| ResNeXt-101-FPN | Motifs | 64.9/66.9 | 15.0/16.4 | 40.8 | 66.8 | 14.1 | 2.5 |
| | **+MBRL** | 56.4/58.3 | **33.7/37.2** | **46.4** | 58.4 | **34.4** | **33.0** |
| | Transformer | 63.5/65.5 | 18.4/20.0 | 41.8 | 65.4 | 19.3 | 6.2 |
| | **+MBRL** | 54.6/56.6 | **32.1/36.1** | **44.8** | 57.6 | **32.6** | **32.6** |
| | VCTree | 64.7/66.6 | 17.2/18.7 | 41.8 | 66.7 | 18.4 | 3.8 |
| | **+MBRL** | 56.4/58.1 | **34.7/38.3** | **46.9** | 58.3 | **34.5** | **35.5** |
| VGG16 | Motifs | 64.4/66.6 | 14.5/16.0 | 40.3 | 66.0 | 13.5 | 2.4 |
| | **+MBRL** | 56.3/58.2 | **33.1/37.0** | **46.2** | 58.5 | **34.3** | **32.7** |
| | Transformer | 62.0/64.2 | 15.6/16.9 | 39.7 | 62.7 | 16.1 | 3.2 |
| | **+MBRL** | 54.8/56.8 | **33.8/37.9** | **45.9** | 57.7 | **34.8** | **34.6** |
| | VCTree | 64.8/66.9 | 17.1/18.8 | 41.9 | 66.3 | 17.2 | 5.3 |
| | **+MBRL** | 56.1/57.9 | **33.9/37.5** | **46.3** | 58.8 | **33.7** | **34.3** |

attributed to MBRL's emphasis on learning fine-grained and infrequent relations. This trade-off is intentional: our method aims to distribute the model's learning capacity more evenly across all relations, rather than overfitting to the head relations that dominate R@100 scores. As a result, our approach effectively mitigates the common bias towards head relations, leading to a more balanced and comprehensive scene graph generation. Moreover, our method can also be adapted to different baseline models with various object detector backbones, with their PredCls results reported in Table 3. We have applied our method to three popular baseline models: motifs, Transformer, and VCTree. The baseline models feature various architectural designs: Motifs utilizes the conventional LSTM structure, VCTree utilizes a tree structure, and Transformer utilizes self-attention layers. Additionally, VCTree combines both reinforcement learning and supervised training. Despite the diversity in model architectures and training methods, our method consistently enhances all models' performance on the mR@50/100 and the mean metrics. The main cause is that through our proposed MBRL, the performance of body and tail relations is significantly enhanced, while the performance of head relations experiences fewer drops.

### GQA200

We also applied MBRL to the more complex GQA200 dataset, as shown in Table 4. From the results, it is validated that MBRL significantly enhances the model's performance on the mR@K metric while keeping the reductions in R@K scores relatively modest, resulting in optimal overall performance on the mean metric. For example, the mean scores of Motifs+MBRL for the three tasks are 44.7, 22.7, and 20.5, respectively. This proves the generalization capabilities of MBRL across various data distributions.

**Table 4 Performance (%) comparison of different methods on the GQA200 dataset.** Bold entries indicate the best results.

| Model | PredCls | | | SGCls | | | SGdet | | |
|---|---|---|---|---|---|---|---|---|---|
| | R@50/100 | mR@50/100 | Mean | R@50/100 | mR@50/100 | Mean | R@50/100 | mR@50/100 | Mean |
| SHA+GCL | 42.7/44.5 | 41.0/42.7 | 42.7 | 21.4/22.2 | 20.6/21.3 | 21.4 | 14.8/17.9 | 17.8/20.1 | 17.7 |
| VTransE | 55.7/57.9 | 14.0/15.0 | 35.7 | 33.4/34.2 | 8.1/8.7 | 21.1 | 27.2/30.7 | 5.8/6.6 | 17.6 |
| VCTree | 63.8/65.7 | 16.6/17.4 | 40.9 | 34.1/34.8 | 7.9/8.3 | 21.3 | 28.3/31.9 | 6.5/7.4 | 18.5 |
| Motifs | 65.3/66.8 | 16.4/17.1 | 41.4 | 34.2/34.9 | 8.2/8.6 | 21.5 | 28.9/33.1 | 6.4/7.7 | 19.0 |
| +GCL | 44.5/46.2 | 36.7/38.1 | 41.4 | 23.2/24.0 | 17.3/18.1 | 20.7 | 18.5/21.8 | 16.8/18.8 | 19.0 |
| **+MBRL** | 55.5/57.2 | 31.9/33.9 | **44.7** | 28.8/29.6 | 15.9/16.6 | **22.7** | 25.0/28.7 | 12.6/15.8 | **20.5** |

**Table 5 Ablation studies on each component of MBRL.** Bold entries indicate the best results.

| MRL | BRL | PredCls | | | | | |
|---|---|---|---|---|---|---|---|
| | | R@50/100 | mR@50/100 | Mean | Head mR@100 | Body mR@100 | Tail mR@100 |
| | | 64.9/66.9 | 15.0/16.4 | 40.8 | 66.8 | 14.1 | 2.5 |
| ✓ | | 58.5/60.3 | 30.7/33.9 | 45.9 | 60.0 | 33.3 | 26.2 |
| ✓ | ✓ | 56.4/58.3 | **33.7/37.2** | **46.4** | 58.4 | **34.4** | **33.0** |

## Ablation studies

MBRL consists of two components: Mixup relation learning (MRL) and balanced relation learning (BRL). As shown in Table 5, we evaluate the impacts of each component of MBRL, which is based on Motifs, in the PredCls task on the VG150 dataset. From the results, we observe that MRL significantly improves the performance in terms of mR@50/100 metric and mean metrics. This demonstrates the effectiveness of MRL in accurately classifying certain coarse-grained relations into their corresponding fine-grained ones. Furthermore, BRL contributes more significantly to improvements in Tail R@100 metric compared to MRL, indicating that BRL plays a crucial role in predicting diverse tail relations. This method effectively protects the learning of tail relation samples and reduces the impact on head relation samples.

## Hyperparameter analysis

### Influence of $\theta$

We investigate the impact of different thresholds $\theta$, ranging from 0.75 to 1, on model performance. As shown in Fig. 4, the R@100 metric shows an increasing trend as the value of $\theta$ increases. Before $\theta$ reaches 0.95, the mR@100 metric remains relatively stable, suggesting that the model maintains consistent performance across tail relations. Once $\theta$ exceeds 0.95, the mR@100 metric significantly decreases. Therefore, based on the results of the mean metrics, we select 0.95 as the optimal threshold.

### Influence of $\alpha$

We experiment with different $\alpha$ values from 2 to 9 to assess the effect of the loss balancing hyperparameter on the model's performance. As shown in Fig. 5, an increase in the value

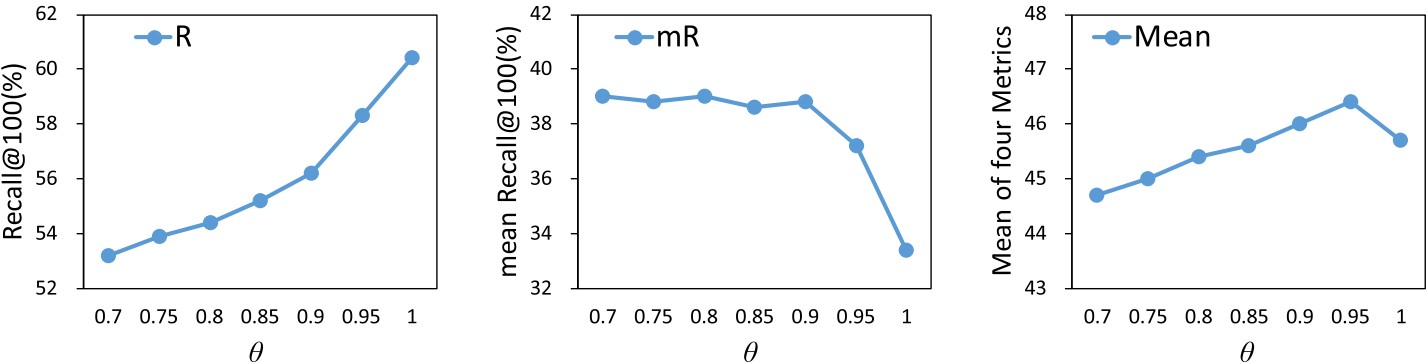

**Figure 4 Influence of θ on our method.** The results are based on the use of motifs for the PredCls task on the VG150 dataset.

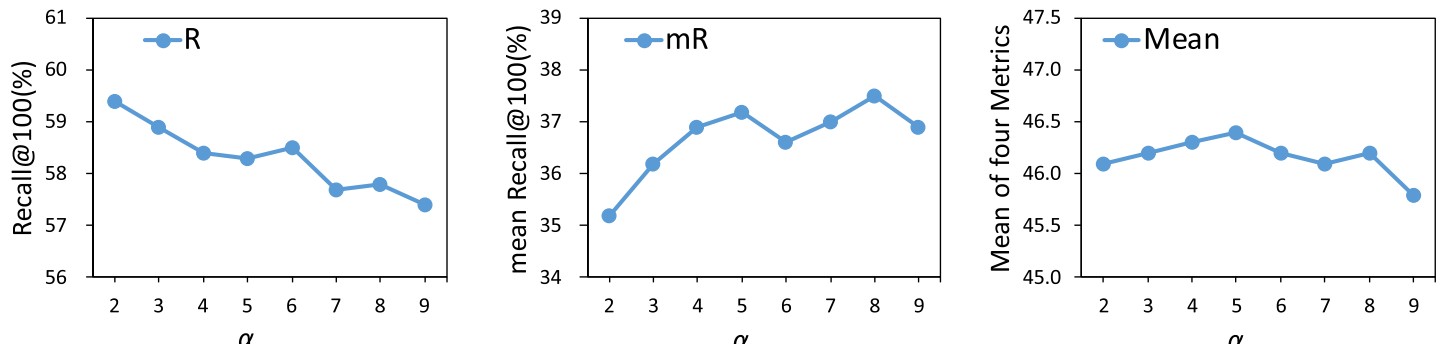

**Figure 5 Influence of α on our method.** The results are based on the use of motifs for the PredCls task on the VG150 dataset.

of $\alpha$ results in a decline in the performance of head relations, while simultaneously enhancing the performance of tail relations. Once the hyperparameter $\alpha$ exceeds 5, the model begins to overfit on tail relations, resulting in diminishing performance gains for these categories. Therefore, based on the mean metric results, the optimal value for $\alpha$ is determined to be 5.

## Visualization results

To demonstrate the effectiveness of our proposed MBRL in accurately identifying relations, we visualize several PredCls examples generated from motifs (with a purple background) and motifs combined with our proposed MBRL (with a blue background) in Fig. 6. Comparing the results of the Motifs, we find that our method can detect more fine-grained relations, such as "*walking on*", "*eating*", "*growing on*", and "*laying on*". MBRL effectively mitigates ambiguity issues and reduces prediction errors in relation recognition by enabling the model to discern subtle differences among relations. Thus, over-confident predictions of head relations under a long-tail distribution can be alleviated to some extent. To illustrate the discriminatory capabilities of MBRL against semantically similar relations, we present

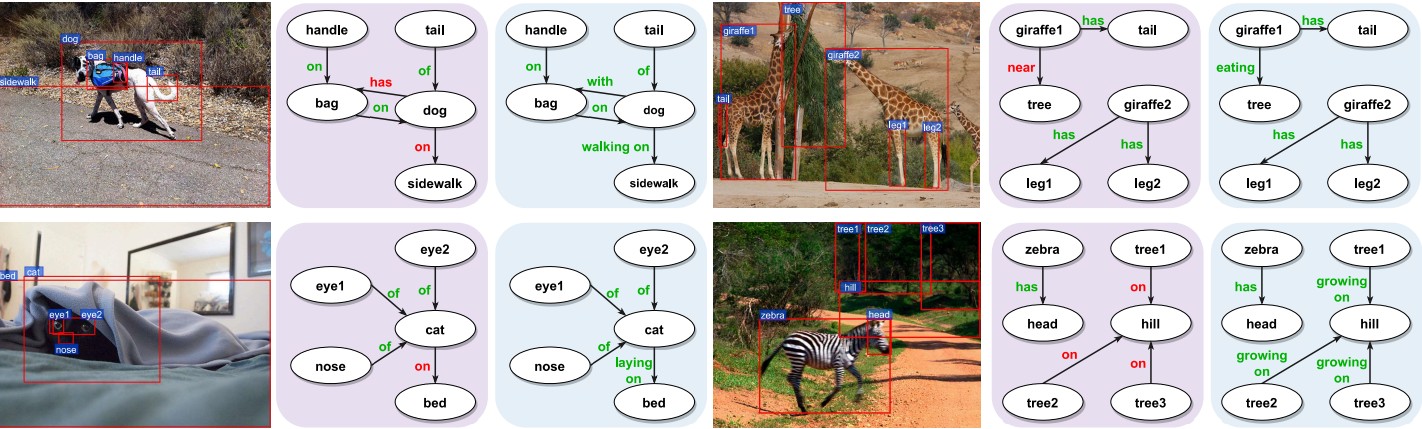

**Figure 6 Visualization results of motifs (with a purple background) and motifs + MBRL (with a blue background) for the PredCls task.** Relations colored in red represent errors, meaning they are not ground-truth relations. Conversely, relations colored in green are correct, indicating that they match the ground-truth relations. Image credit: the Visual Genome dataset archive at https://homes.cs.washington.edu/~ranjay/visualgenome/.

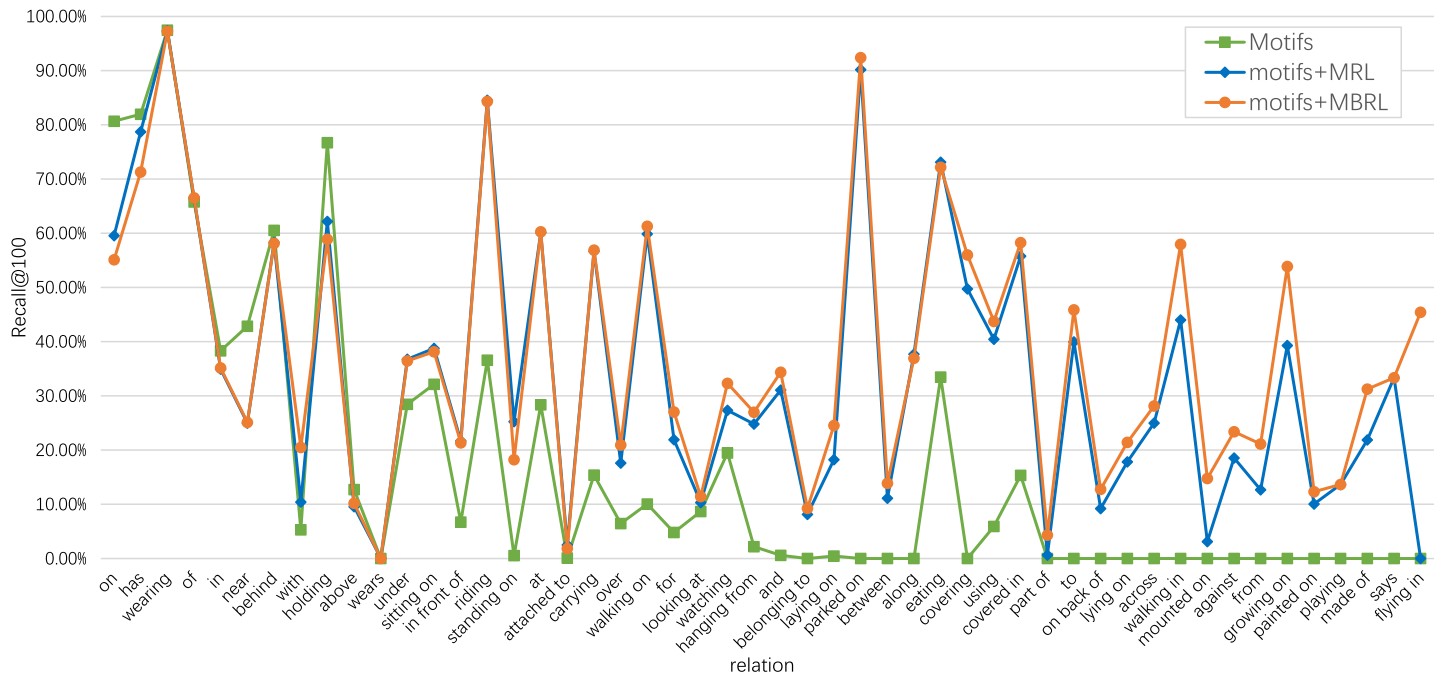

**Figure 7 Comparison of Recall@100 among motifs, motifs+MRL, and motifs+MBRL for each relation category of the PredCls task on the VG150 dataset.** The frequencies of relations decrease from left to right.

the PredCls results of Motifs+MBRL in Fig. 7. Observations indicate that Motifs+MRL leads to enhancements in most relations. However, for challenging predictions, such as "*flying in*" and "*mounted on*", Motifs+MRL is susceptible to errors due to the long-tail distribution. Conversely, BRL significantly bolsters the model's ability to distinguish between fine-

grained and infrequent relations. These results demonstrate that our proposed MBRL can enhance scene graph generation by generating more reasonable relations.

## CONCLUSION

In this article, we introduce the MBRL framework designed to mitigate semantic ambiguity and address the long-tail distribution challenges in SGG. Our method enhances the training data by assigning soft labels to samples with semantic ambiguity and optimizes model performance through adjustment of loss weights for fine-grained and low-frequency relation samples. MBRL effectively mitigates the bias towards frequently occurring but less informative relations. Moreover, the model-agnostic design of MBRL allows seamless integration with various SGG architectures, including motifs, Transformer, and VCTree, independent of their underlying object detector backbones. However, MBRL focuses primarily on relation prediction and does not directly address imbalances in object category distributions, which could affect overall scene understanding. To overcome these limitations, future work will extend MBRL to address object category imbalances, aiming for robustness in both object detection and relation prediction under long-tail distributions. Finally, we plan to explore the application of MBRL in downstream tasks, such as image caption generation and visual question answering, to further demonstrate its versatility.

## ACKNOWLEDGEMENTS

We used the Visual Genome dataset, created by Ranjay Krishna, and the Generalized Question Answering dataset, created by Drew A. Hudson and Christopher D. Manning. Both datasets are licensed under a Creative Commons Attribution 4.0 International License. The original Visual Genome dataset is available at https://homes.cs.washington.edu/~ranjay/visualgenome/, and the original Generalized Question Answering dataset is available at https://cs.stanford.edu/people/dorarad/gqa/. We are deeply grateful to the authors of these datasets for their efforts in advancing the field of Scene Graph Generation by making these resources publicly available.

### Funding

This work was supported by the Key-Area Research and Development Program of Guangdong Province under Grant (2019B111101001) and the Science and Technology on Information System Engineering Laboratory (WDZC20205250410). The funders had no role in study design, data collection and analysis, decision to publish, or preparation of the manuscript.

### Grant Disclosures

The following grant information was disclosed by the authors:
Key-Area Research and Development Program of Guangdong Province: 2019B111101001.

Science and Technology on Information System Engineering Laboratory: WDZC20205250410.

## Competing Interests

The authors declare that they have no competing interests.

## Author Contributions

- Shanjin Zhong conceived and designed the experiments, performed the experiments, analyzed the data, performed the computation work, prepared figures and/or tables, authored or reviewed drafts of the article, and approved the final draft.
- Yang Cao conceived and designed the experiments, performed the experiments, prepared figures and/or tables, authored or reviewed drafts of the article, and approved the final draft.
- Qiaosen Chen conceived and designed the experiments, prepared figures and/or tables, and approved the final draft.
- Jie Gong analyzed the data, prepared figures and/or tables, and approved the final draft.

## Data Availability

The Visual Genome dataset is available at https://homes.cs.washington.edu/~ranjay/visualgenome.

The Generalized Question Answering dataset is available at https://cs.stanford.edu/people/dorarad/gqa.

The code is available at GitHub and Zenodo:

- https://github.com/ZhongShanjin/MBRL-master.

- ShanJin Zhong. (2024). ZhongShanjin/MBRL-master: v1.0.0 (v1.0.0). Zenodo. https://doi.org/10.5281/zenodo.14442298.

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
