# Peer review of "Learning with semantic ambiguity for unbiased scene graph generation"

_PeerJ Computer Science, doi:10.7717/peerj-cs.2639_

## Round 0.1 · original submission · Major Revisions

Dear authors,

Thank you for the submission. The reviewers’ comments are now available. It is not suggested that your article be published in its current format. We do, however, advise you to revise the paper in light of the reviewers’ comments and concerns before resubmitting it. The followings should also be addressed:

1. Information about the datasets should be provided in the Abstract section.
2. Many of the equations are part of the related sentences. Attention is needed for correct sentence formation.
3. Equations should be used with correct equation number. Please do not use “as follows”, “given as”, etc. Explanation of the equations should also be checked. All variables should be written in italic as in the equations. Their definitions and boundaries should be defined. Necessary references should be provided.
4. All of the values for the parameters of all algorithms should be given.
5. Many of the equations are part of the related sentences. Attention is needed for correct sentence formation.
6. Some more recommendations and conclusions should be discussed about the paper considering the experimental results. The conclusion section is weak. There is also no discussion section about the results. It should briefly describe the results of the study and some more directions for further research. You should describe the academic implications, main findings, shortcomings and directions for future research in the conclusion section. The conclusion in its current form is generally confused. What will be happen next? What we supposed to expect from the future papers? So rewrite it and consider the following comments:
- Highlight your analysis and reflect only the important points for the whole paper.
- Mention the benefits
- Mention the implication in the last of this section.

Best wishes,

Reviewer 1 ·

Basic reporting

The paper fails to introduce the background of scene graph generation. In the introduction, it would be better to explain what are the nodes and edges in a scene graph and what are the objectives in scene graph generation. It only becomes clear at Line 94.

Abuse of notation. Line 179, "s" in $r_j^s$ represents soft label while in Eq. (1), "s" denotes a subject.

In Eq. (5), it is unclear what is the soft label $r_*^s$. Based on the illustration in Fig. 2, $r_*^s$ changes for different relation categories. Moreover, it seems that the loss in Eq. (5) is only the per-sample loss. It is suggested to clearly show the overall optimization objective.

Experimental design

The problem setting is not clear. Scene graph generation, based on the description in the paper, involves identifying objects and relations. However, in the method section, the focus shifts to only subject-object relation classification. It is suggested to clearly describe the problem setting and the task to solve.

On Line 211, the authors mention three kinds of tasks with different prediction targets. Given that the method mainly focuses on relation prediction, how can the proposed method be applied to these three tasks? What if the object categories also have imbalanced distributions?

Validity of the findings

Based on the results in Table 1, it is not reasonable to claim that MBRL shows minimal performance degradation on the R@K metric. For example, Motifs achieves a R@100 of 66.9 on PredCIs. But with MBRL, the metric reduces to 58.3, which is a significant drop. It is suggested to explain this phenomenon in detail.

Additional comments

When calculating the R@K metric, what are considered as positive samples?

The paper states that in the MRL module, samples having low-frequency relations are not assigned soft labels (three lines above Eq. (7)). This is not true. As stated in Line 183, soft labels are assigned to all samples in $T_sim$, which by definition, is the set of all misclassified samples. Samples with low-frequency relations may be misclassified as the model may be biased towards predicting high-frequency relations. It would be better to clarify this point.

Cite this review as

Reviewer 2 ·

Basic reporting

This paper tackles important issues in Scene Graph Generation algorithms, that often make them hard to be use for downstream tasks, namely the long tail distribution of relationships among categories, and semantic ambiguity of relationships among two objects. To tackle these issues, a framework called Mixup and Balanced Relation Learning (MBRL) is proposed. MBL is model agnostic and is proposed to be seamlessly integrated in SGG models. MBRL consists in first creating soft labels for relationships among two objects, and then select fine grained labels according to a threshold. This allows considering not only one label for a specific triple but multiple. The second step is for those low frequency relationships, which do not have fine grained relationships, a lose balancing hyperparameter is used to adjust the loss weight of such samples.

The paper is well written, well-structured and presents a good understanding of the issue and the framework proposed.

Experimental design

This contribution presents a set of experiments, namely on tasks (e.g, predicate classification, scene graph classification, and scene graph detection), ablation studies, and hyperparameter analysis. To make the experiments comprehensive, what I am missing are performance metrics, and comparison with other methods in this regard.

Validity of the findings

The experiments results presented in this contribution show that MBRL is successful and competitive with current state of the art algorithms. For this contribution to be used and improved on by the community, as well as to be able to reproduce the results, authors should make available their source code, with clear instructions on how to reproduce it. I know that the submission includes the code, but not as a repository but as a link to a google drive.

I am looking forward to the future work on expanding the method for downstream applications.

Cite this review as

---

## Round 0.2 · accepted · Accept

Dear Authors,

Thank you for addressing the reviewers" comments. Your manuscript seems now ready for publication.

Best wishes,

Reviewer 1 ·

Basic reporting

no comment

Experimental design

no comment

Validity of the findings

no comment

Additional comments

Thanks to the authors for addressing my concerns and thoroughly revising the paper.
The current version significantly improves over the previous one in terms of clarity.

Cite this review as